# Production and Processing of a Spherical Polybutylene Terephthalate Powder for Laser Sintering

**Rob G. Kleijnen** [1,*] , **Manfred Schmid** [1] **and Konrad Wegener** [2]

[1]  inspire AG, Innovation Center for Additive Manufacturing Switzerland, 9014 St. Gallen, Switzerland;
    manfred.schmid@inspire.ethz.ch
[2]  Swiss Federal Institute of Technology, ETH Zurich, Institute of Machine Tools and Manufacturing,
    8092 Zürich, Switzerland; wegener@iwf.mavt.ethz.ch
[*]  Correspondence: kleijnen@inspire.ethz.ch



**Featured Application: This work describes in detail a process chain for the production of spherical laser sintering powders. The shown methods can be applied to extend the number of suitable laser sintering materials.**

**Abstract:** This work describes the production of a spherical polybutylene terephthalate (PBT) powder and its processing with selective laser sintering (SLS). The powder was produced via melt emulsification, a continuous extrusion-based process. PBT was melt blended with polyethylene glycol (PEG), creating an emulsion of spherical PBT droplets in a PEG matrix. Powder could be extracted after dissolving the PEG matrix phase in water. The extrusion settings were adjusted to optimize the size and yield of PBT particles. After classification, 79 vol. % of particles fell within a range of 10–100 μm. Owing to its spherical shape, the powder exhibited excellent flowability and packing properties. After powder production, the width of the thermal processing (sintering) window was reduced by 7.6 °C. Processing of the powder on a laser sintering machine was only possible with difficulties. The parts exhibited mechanical properties inferior to injection-molded specimens. The main reason lied in the PBT being prone to thermal degradation and hydrolysis during the powder production process. Melt emulsification in general is a process well suited to produce a large variety of SLS powders with exceptional flowability.

**Keywords:** selective laser sintering; powder bed fusion; additive manufacturing; PBT; powder; melt emulsification; powder production; polymer processing

---

## 1. Introduction

Industry increasingly adopts additive manufacturing (AM) as a way to produce prototypes and small series of end-use parts in a fast, reliable, and cost-effective manner. Multiple studies [1–3] indicate that amongst the AM technologies that use polymers as a feedstock, laser sintering offers high levels of design freedom and productivity. The limitations of the laser sintering process are the high equipment investment costs, and the limited number of usable materials.

Laser sintering is a powder-based process wherein parts are created layer-by-layer. A roller or blade first deposits a thin powder layer of typically 100–150 μm thickness. The cross-sections of parts present in this layer are molten by scanning the surface with a $CO_2$ laser. As a result, the powder particles fuse together with the underlying layer and with each other. Once scanning has finished, the build platform lowers by a single layer thickness, and the process is repeated. In this way, three-dimensional objects are created.

A powder material must fulfill a number of requirements in order to be suitable for laser sintering, as summarized by Schmid [4]. Materials must be carefully designed, both on an intrinsic polymer level, and on an extrinsic powder level. The most important requirements are summarized in Figure 1.

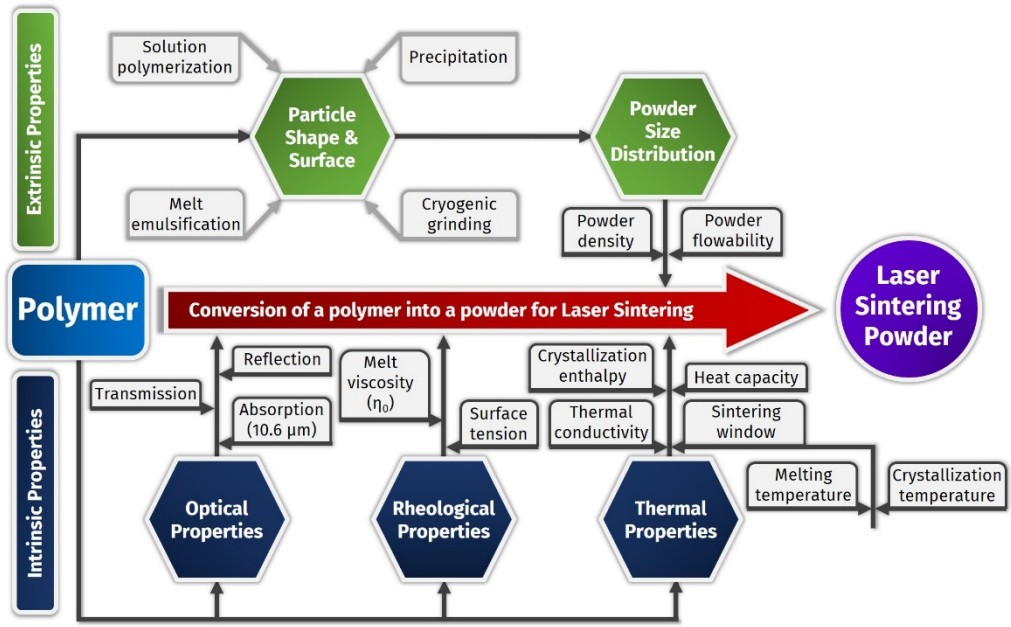

**Figure 1.** Critical properties for a polymer laser sintering material. Adapted with permission from Schmid et al. [5].

The intrinsic polymer properties in the bottom section of the diagram in Figure 1 can be established by choice of the material grade, or better, by specific design of the polymer chains and architecture via chemistry. The extrinsic properties in the top part of the diagram are those associated with the shape, size, and mechanics of the powder particles the material consists of. Tuning and optimization of these properties is done by selection of an appropriate powder production technology and subsequent classification methods.

Because of these many requirements, there are only a handful materials on the market that can be used with the laser sintering process. Market studies [6] show that the largest portion of parts built with laser sintering is based on polyamide (PA), but that there is a strong desire for parts made from different materials. One of these materials is polybutylene terephthalate (PBT). PBT has a good chemical resistance and is characterized by a high insulation resistance and dielectric strength. This makes it the material of choice for housings for electrical appliances and various under-the-hood car parts. The ability to produce such parts with laser sintering, thereby making use of the full design freedom the process offers, opens up many possibilities. Examples are weight reduction, downsizing, and consolidation of many parts into a single laser sintered one.

Various authors described the production and use of PBT powders for laser sintering. Investigating five different PBT grades, Wegner et al. [7] reported the production of powders by cryogenic grinding, and their subsequent processing on an SLS machine. The parts exhibited a relatively high Young's modulus. However, tensile strength and elongation at break compared to injection-molded parts were limited, signaling a need for further improvement.

Schmidt et al. [8–11] produced powders by wet grinding PBT granules, then running the milled powder through a so-called downer reactor to obtain spherical particles. Parts consisting only of a single layer could be produced with laser sintering, but no mechanical properties were tested. The performance of dry blends of PA12 mixed with up to 30% PBT was investigated by Salmoria et al. [12]. The authors reported a minimal increase of flexural modulus upon blending PA12 with 10% PBT. The

blends were otherwise featured by phase separation and inferior mechanical properties compared to the PA12 base component.

The mechanical properties and crystalline structure of laser-sintered parts made from a cryogenically milled PBT copolymer powder were reported by Arai et al. [13]. The copolymer, which contained 10 mol % isophthalic acid as a comonomer, was processed on a laser sintering machine at 190 °C. The produced parts exhibited only a slightly reduced strength compared to their injection-molded counterparts, but a drastically reduced elongation at break. In a subsequent study [14], the same authors investigated the behavior of PBT powder dry blended with short glass fibers.

Most of the work up to this point was carried out on PBT produced by grinding granules into powder, typically resulting in irregularly shaped particles that give a powder suboptimal flowability and packing density. In this article, melt emulsification is used as an alternative method to directly create PBT powder consisting of spherical particles.

Various researchers showed the advantages of using powders with spherical particles in laser sintering. Berretta et al. [15,16] characterized different powders with respect to size distribution, and particularly particle shape. The flowability as expressed by measurement of the angle of repose was found to correlate with the circularity and roundness of powder particles. Verbelen [17] provided screening methodologies for laser sintering powders, which particularly included assessment of the powder flowability. Van den Eynde [18], following this methodology, extensively characterized the relations between size, shape, flowability, and performance of a wide range of materials. It was found that in general, a smooth and spherical particle shape contributes to the laser sintering material performance. In a comparison of two laser-sinterable PA12 powders, Schmid et al. [19] found increased tensile strength and elongation at break for the powder with the higher sphericity. This was attributed at least partially to the improved packing density achieved with spherical particles. Quantitative results relating a more spherical particle shape to improved flowability and laser sintering performance were presented by Amado [20] and Vetterli et al. [21].

The term melt emulsification refers to the process of melt blending two incompatible materials to create a structure consisting of droplets of one component embedded in the matrix of the other component. The melt emulsification method is well-established and reviewed [22] in the pharmaceutical domain to produce dispersions of active components with spherical morphology in a polymer matrix. For the production of laser sintering powders, the method is less well documented. Drummer et al. [23] reported the production of spherical PA12 particles by melt blending with polyvinyl alcohol (PVA). The authors also showed some preliminary results on blends of PBT and polyethylene glycol (PEG). The production of spherical polypropylene particles via melt blending with hexadecane was described by Fanselow et al. [24,25], however no parts were produced with the obtained powder.

The current work describes the production of a PBT powder consisting of spherical particles by melt emulsification with PEG as the matrix phase. The advantage of this process route is the absence of organic solvents and the requirement of only elementary polymer processing equipment; as will be demonstrated, it is already possible to produce reasonable amounts of powder that allow for laser sintering trials using a conventional laboratory-scale single-screw extruder.

In an emulsion, the equilibrium between break-up and coalescence governs the mean droplet size. For a system in shear, the droplet break-up depends on the shear rate and relative viscosity of the two phases. The coalescence is mainly governed by interfacial tension, which determines the probability of recombination of two colliding droplets. A system consisting of non-Newtonian materials flowing in a non-uniform shear field across a wide temperature range, as is the case in a typical extruder, cannot be described by simple models. Attempts have been made to describe the droplet size as a function of material and process parameters, as has been reviewed by multiple authors [26,27]. These models however typically have an empirical basis and hold only for the particular investigated system.

As a first step in the current investigation, the extruder rotational speed and nozzle were varied, and the resulting particle size was evaluated. The influence of blend ratio PBT/PEG on the particle size was investigated as well. An ideal and aimed for size range for typical laser sintering powders

lies between 10–100 μm [19]. After optimization of extruder settings and blend ratio, the powder was extracted from the extrudate through a series of washing, drying, and classification steps. These steps are summarized visually in Figure 2. The particle size and shape distribution, thermal properties, and flowability properties of the resulting powder were analyzed. The material was then tested on a commercial laser sintering machine with specifically designed inserts to allow for processing with small amounts of powder. Finally, the mechanical properties of the built parts were evaluated and compared to those of compression and injection molded samples made from the same PBT grade.

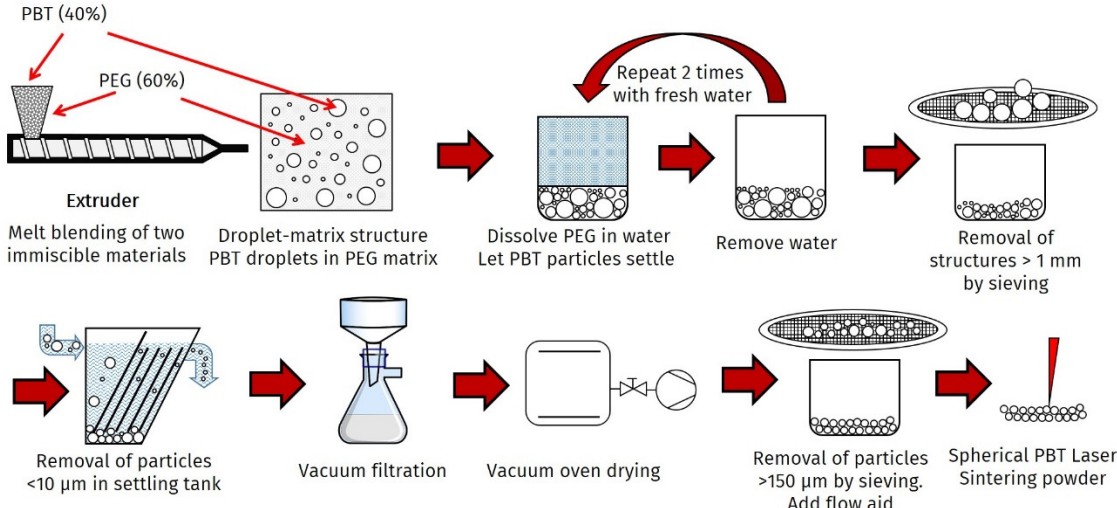

**Figure 2.** Schematic overview of the powder production process. PBT: polybutylene terephthalate.

## 2. Materials and Methods

Materials: PEG Polyglykol 35000 S (Clariant, Muttenz, Switzerland) flakes were ordered and used without further purification or drying. PBT TORAYCON 1200M (TORAY, Tokyo, Japan) was purchased in granulate form. The material did not contain any stabilizers or other additives so as to prevent these from negatively influencing the laser sintering process. The material was dried before use. A second reference PBT CCP PBT 1100-S600 (Chang Chung Plastics, Taipei, Taiwan) granulate was acquired as well. This material contained stabilizers and process aids. The material was dried before use. Fumed silica flow aid Aerosil R812 (Evonik, Essen, Germany) used without further alteration.

Extrusion: the materials were mixed in a laboratory scale Extrusiograph (Brabender Technologie, Duisburg, Germany) single-screw extruder with a barrel temperature profile of 230-240-250-250 °C going from inlet to nozzle. The screw had a 19 mm base diameter and L/D ratio of 25.7. A wide-slit nozzle was used to produce the bulk of the material, which was collected and passively cooled down against ambient air on a steel sheet.

Powder extraction: the slabs obtained from extrusion were broken into small pieces and divided into batches of 2.5 kg each. To each batch, 10 L water was added and stirred in a container inside a concrete mixer for two hours. The mixture was then allowed to settle for 12 hours, after which the watery phase was removed. Another 10 L fresh water was added to the remaining slurry, and the process was repeated. The complete washing cycle was carried out three times for each batch. Removal of the PEG phase was confirmed by DSC measurement.

Powder classification and drying: the fine fraction of the powder, consisting of particles smaller than 10 μm, was removed using a specially designed settling tank. Depending on the flow rate of a dilute dispersion of powder in water through the settling tank, particles of certain size settle to its bottom. Small enough particles with reduced settling velocity do not reach the bottom of the tank before the water reaches the outlet, and are washed out in this way. In a subsequent step, the slurry at the bottom of the tank was collected, and the majority of water was removed by vacuum filtration over a Büchner funnel. The final traces of water were removed by drying the material for 24 hours at

80 °C in a vacuum drying oven. The powder was then sieved over a sieve with a 150 μm mesh width. The powder was dry blended with 0.05 wt. % fumed silica as flowing aid.

Scanning electron microscopy: scanning electron microscopy images were obtained on a JSM 7100F scanning electron microscope (JEOL, Tokyo, Japan). Samples were deposited on a piece of carbon tape on an aluminium SEM stub and sputter coated with a 10 nm layer of Pt/Pd (80/20).

Optical microscopy: to assess the morphology and microstructure of built parts, a sample was fractured and embedded in resin. The sample was sanded, polished, and observed using a DM6 optical microscope (Leica Microsystems, Wetzlar, Germany) with incident light and differential interference contrast.

Particle size distribution: the particle size distribution was measured by dynamic light scattering on an LS230 (Beckman Coulter, Brea, CA, United States) instrument. Approximately 0.1 g of powder was added to approximately 20 ml demineralized water and thoroughly stirred. Before measurement, the dispersion was treated with ultrasound to break apart any agglomerates. The particle size was calculated based on the Fraunhofer model, for the range between 0.4–2000 μm.

Particle size and shape distribution: the size and shape of particles was recorded optically with a DM6 optical microscope (Leica Microsystems, Wetzlar, Germany) at 100x magnification in transmission. On the basis of 216 images, 51862 particles were counted and evaluated using an in-house developed imageJ script and MATLAB evaluation procedure, described in [28].

Powder flowability: characterization of the powder flowability was carried out on a Revolution Powder Analyzer (RPA) (Mercury Scientific, Newtown, CT, United States). An exact measure of 25 mL powder at tapped density was added to a rotating drum with a diameter of 50 mm. The drum was rotated with a speed of 0.6 rpm, while a camera recorded 384 avalanche events. Each time directly following an avalanche, the avalanche angle and roughness of the powder surface were evaluated.

Thermal analysis: differential Scanning Calorimetry (DSC) measurements were carried out on a DSC 25 (TA Instruments, New Castle, DE, United States). All measurements were conducted under a nitrogen atmosphere, from 25 °C to 250 °C, with heating and cooling rates of 10 °C/min.

Rheology: a RG20 high pressure capillary rheometer (Göttfert, Buchen, Germany) was used to assess rheological properties. Measurements were carried out at 250 °C with a nozzle with a 1 mm diameter and 20 mm length. Both channels were used simultaneously for measurement, only relative results are provided.

Mechanical tests: mechanical tests were performed on a Z100 universal testing machine (Zwick Roell, Ulm, Germany). The tests were performed according to the ISO 527-1 standard, with testing geometries of type ISO 527-2A-1BA-25.

Laser Sintering: laser sintering trials were conducted on a Sinterstation 2000 commercial laser sintering machine (DTM, Austin, TX, United States) equipped with inserts on both sides of the central powder bed to allow for the processing of reduced amount of material. The machine disposes of a 50 W $CO_2$ laser operating at a wavelength of 10.6 μm. New layers were deposited on the powder bed by means of a counter-rotating roller. The process parameters are listed in Table 1 below.

**Table 1.** Laser sintering process parameters.

| Parameter | Symbol | Value |
|---|---|---|
| Powder bed temperature | $T_{bed}$ | 213–215 °C |
| Piston temperature | $T_{piston}$ | 160 °C |
| Cylinder temperature | $T_{cylinder}$ | 200 °C |
| Powder feed temperature | $T_{feed}$ | 170 °C |
| Laser power | $P_L$ | 8 W |
| Laser scan speed | $v_s$ | 5 m/s |
| Hatch distance | $d_s$ | 0.30 mm |
| Layer thickness | $t_l$ | 0.12 mm |
| Energy density | $\rho_E$ | 44.4 J/cm$^3$ |

## 3. Results and Discussion

### 3.1. Extrusion

The optimal particle size range of polymer powders used for laser sintering lies between 10 and 100 μm. In order to obtain a size distribution that approaches this ideal range as close as possible, in total 65 trials with different extruder parameters were executed. Amongst the varied parameters were material grade selection, mixing ratio, extruder rotational speed, and nozzle selection. In this paper, only the results of varying rotational speed and nozzle will be shown, to specifically demonstrate the influence of extruder parameters on the particle size. A reduction of mean particle size is expected with higher rotational speeds. The choice of nozzle influences the amount of pressure built up within the extruder, and with this the shear forces the polymer blend is subjected to. Four types of nozzles were used, three of which had a cylindrical geometry. The fourth nozzle was a 2 × 100 mm wide slit that caused by far the least amount of back pressure in the extruder. The settings used for selected trials are listed in Table 2.

**Table 2.** Extruder settings for selected extrusion trials. PBT: polybutylene terephthalate; PEG: polyethylene glycol.

| Trial # | Speed [rpm] | Nozzle | PBT Content [wt. %] | PEG Content [wt. %] |
|---------|-------------|--------|---------------------|---------------------|
| 1 | 30 | ⌀ = 1 mm; l = 20 mm | 40 | 60 |
| 2 | 60 | ⌀ = 1 mm; l = 20 mm | 40 | 60 |
| 3 | 30 | ⌀ = 3 mm; l = 20 mm | 40 | 60 |
| 4 | 60 | ⌀ = 3 mm; l = 20 mm | 40 | 60 |
| 5 | 30 | ⌀ = 3 mm; l = 10 mm | 40 | 60 |
| 6 | 60 | ⌀ = 3 mm; l = 10 mm | 40 | 60 |
| 7 | 30 | Wide slit nozzle (WSN) | 40 | 60 |
| 8 | 60 | Wide slit nozzle (WSN) | 40 | 60 |

The particle sizes measured by dynamic light scattering are summarized in two different ways in Figures 3 and 4a. Figure 3 shows the evolution of the $D_{10}$, $D_{50}$, and $D_{90}$. These numbers represent the diameter of particles at which exactly 10%, 50%, and 90% of total particle volume is smaller than the respective diameter. The $D_{50}$ gives the median of the distribution, the $D_{10}$ and $D_{90}$ give an indication of the distribution width. In Figure 3, the numbers are shown for each trial, i.e. in dependence of extruder rotational speed (rpm), and nozzle selection. The extruder rotational speed has a clear influence on the particle size; smaller particles are generated at higher speeds, as was expected. At 60 rpm speeds, the nozzle selection only has a minor influence on particle size.

The median value of the particle size ($D_{50}$) lies within the desired range between 10–100 μm, regardless of process parameters. This observation is supplemented by results shown in figure 4a, which shows the percentage of particles in the total volume that falls within the desired size range. Even though the change in extruder screw speed leads to a clear shift in the distributions as seen in Figure 3, the relative number of particles in the desired range fluctuates only by a few percent.

Next to particles, also a certain amount of fibrous PBT is generated during extrusion. These fibers are most likely produced due to insufficient mixing time to achieve fiber break-up. Because the system is relatively robust, the choice of definitive process parameters was mainly aimed at minimizing the amount of fibers. Therefore, the amount of PBT fibers relative to the total PBT content was weighed for each trial.

The generation of fibers strongly depends on the extruder rotation speed, as can be seen in figure 4b. By far the smallest amount of fibrous residue was collected when the wide slit nozzle was used with an extruder rotational speed of 30 rpm. All subsequent powder production trials were conducted with these settings.

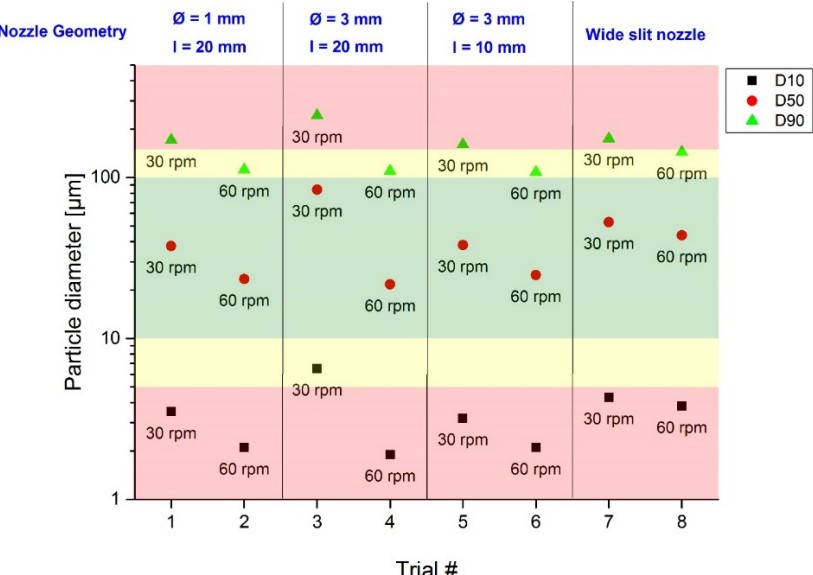

**Figure 3.** Particle size ($D_{10}$, $D_{50}$, $D_{90}$) in dependence of nozzle selection and extruder rotational speed. Colored regions indicate optimal size region (green), suboptimal size regions (yellow), and undesired size regions (red).

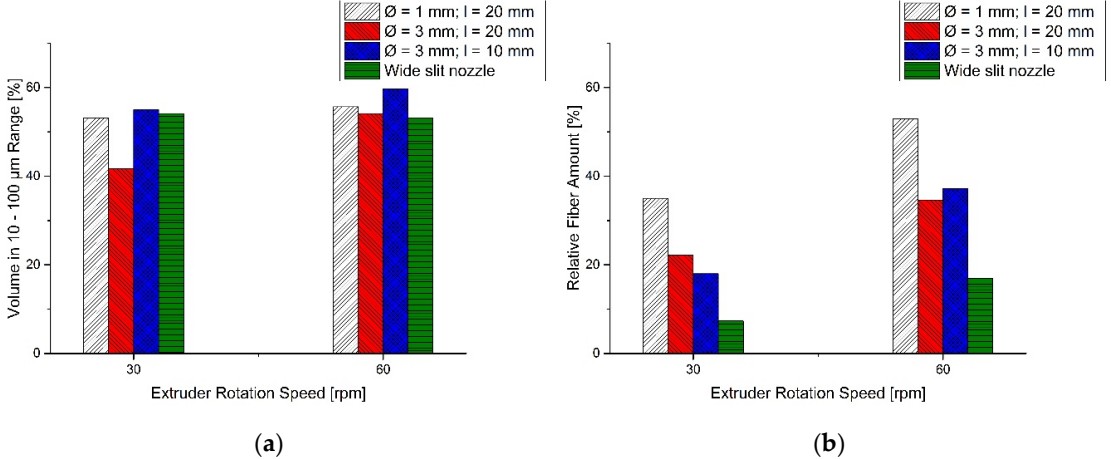

(**a**)　　　　　　　　　　　　　　(**b**)

**Figure 4.** (**a**) Percentage of total particle volume with a size that lies in the range of 10–100 μm in dependence of nozzle selection and extruder rotational speed; (**b**) Amount of PBT that comes out of the extruder in fibrous state, relative to total PBT blend content.

*3.2. Powder classification*

In order to produce a sufficient amount of powder for laser sintering trials, 9 kg PEG flakes were dry blended with 6 kg PBT granules, and extruded using the process parameters from Trial 7, as discussed in the previous section. After removal of the PEG matrix phase, approximately 1.5 kg fibrous mass was extracted. This translates to a conversion efficiency of 75%, going from PBT granulate to PBT powder. The particle size distribution for the as-extruded PBT powder is shown in Figure 5. The distribution is rather broad, and contains a considerable amount of fine particles. Of the total volume, 67% falls within the 10–100 μm range, which is 10% more than measured during initial trials. As shown by van den Eynde [18] and Ziegelmeier et al. [29], who studied the performance of two thermoplastic elastomers with different shapes and size distributions, the reduction of fine particles can improve flowability and lead to a higher packing density. The part properties can be improved in this way.

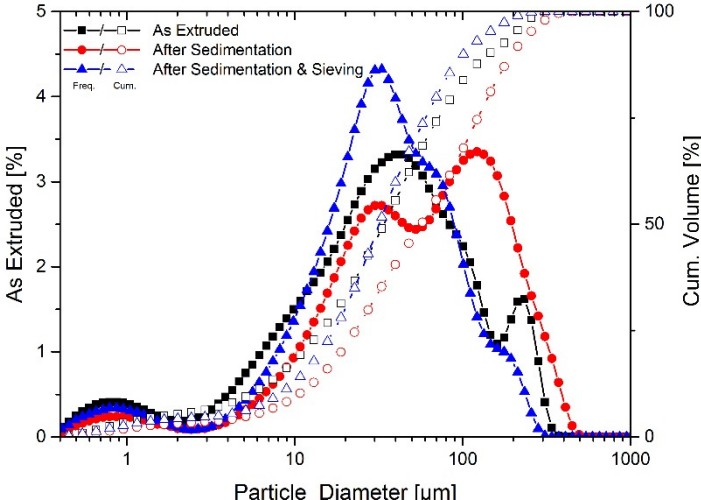

**Figure 5.** Volume-based particle size distributions for the PBT powder after different processing steps: after extrusion, sedimentation in the settling tank, and sedimentation & sieving with a 150 μm sieve. Solid curves and symbols represent the frequency distributions, open symbols with dashed lines represent the cumulative distributions.

To reduce the amount of fine particles in the distribution, the powder was led through a settling tank that was specifically designed and constructed for this purpose. The functional principle is based on Stokes' law, which equates the frictional force between a fluid and a spherical particle to the fluid's viscosity and flow velocity, and the density and size of said particle. The terminal settling speed of a particle can then be determined by balancing the gravitational force on a particle against the frictional force as given by Stokes' law and the buoyancy, and solving for velocity:

$$v_\infty = \frac{\left(\rho_p - \rho_f\right)}{18\mu} gD^2, \tag{1}$$

with $\rho_p$ and $\rho_f$ the densities of particle and fluid respectively, $\mu$ the fluid viscosity, g the gravitational constant, and D the diameter of the particle. For particles with a size of 10 μm, the terminal settling speed was calculated according to (1) to be 0.02 mm/s. The density of the particles was assumed to be equal to the material density of 1.31 g/cm$^3$. The density and dynamic viscosity of water at room temperature were taken from literature [30] and set at 1.0 g/cm$^3$ and 0.89 mPa.s, respectively. The flow rate in the settling tank depends on its geometry. A schematic representation of the settling tank is shown in Figure 6a. The tank has a total volume of 148 liters and is fitted with ten plates inclined at an angle of 60°. The flow rate Q to remove particles with a certain maximum terminal settling velocity for a settling tank with this geometry is given by:

$$Q = v_\infty NA \cos \alpha, \tag{2}$$

with N and A the number of plates and their respective area, and $\alpha$ the inclination angle of the plates. To remove particles smaller than 10 μm, i.e., to prevent them from settling at the bottom of the tank, the terminal settling speed from (1) is plugged into (2). It then follows that a flow rate of 1.4 L/min should be set. To verify that this is the correct flow rate, the particle size in the effluent was measured by dynamic light scattering. The results are shown in Figure 6b, where it becomes clear that the effluent only contains the desired particles smaller than 10 μm. The distribution has a $D_{10}$ of 0.74 μm, $D_{50}$ of 4.24 μm, and $D_{90}$ of 8.43 μm.

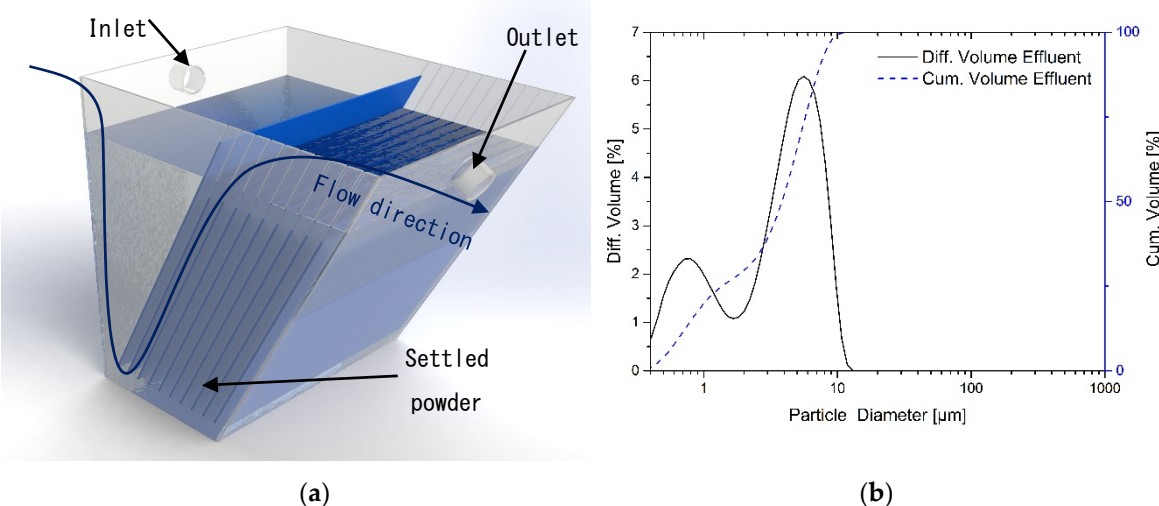

(**a**)          (**b**)

**Figure 6.** (**a**) Schematic representation of the designed settling tank; (**b**) Particle size distribution in the effluent from the outlet of the settling tank.

The fact that fine particles were removed can also be observed in the size distribution of the settled powder left at the bottom of the tank. In Figure 5, this distribution, represented by the red curve, is clearly shifted to the right with respect to the initial, as extruded distribution. To make the powder suitable for laser sintering in terms of particle size, also the largest particles need to be removed. This was done by sieving the powder through a vibrating sieve with a 150 μm mesh width. The final resulting size distribution is also shown in Figure 5 (blue curve). Table 3 lists the characteristic values for the distributions. Through the entire classification process, the percentage of particles within the desired 10–100 μm range can be increased by at least 10, from 67.7 to 78.7.

**Table 3.** Particle size distributions of the powder after different classification process steps.

| Powder | D10 [μm] | D50 [μm] | D90 [μm] | Volume in 10–100 μm Range [%] |
|---|---|---|---|---|
| As-extruded (30 rpm, WSN) | 5.95 | 37.33 | 156.9 | 67.7 |
| After settling tank | 11.58 | 62.45 | 217.6 | 59.6 |
| After settling tank & sieving | 8.87 | 35.0 | 111.0 | 78.71 |

*3.3. Particle Morphology and Shape*

SEM images of the produced powder are shown in Figure 7a,b. The powder has a spherical morphology with smooth surfaces. The particles are solid and do not show any porosity on the surface. Despite improvements to the size distribution by classification, particles smaller than 10 μm are still visible, often clustered together against larger particles.

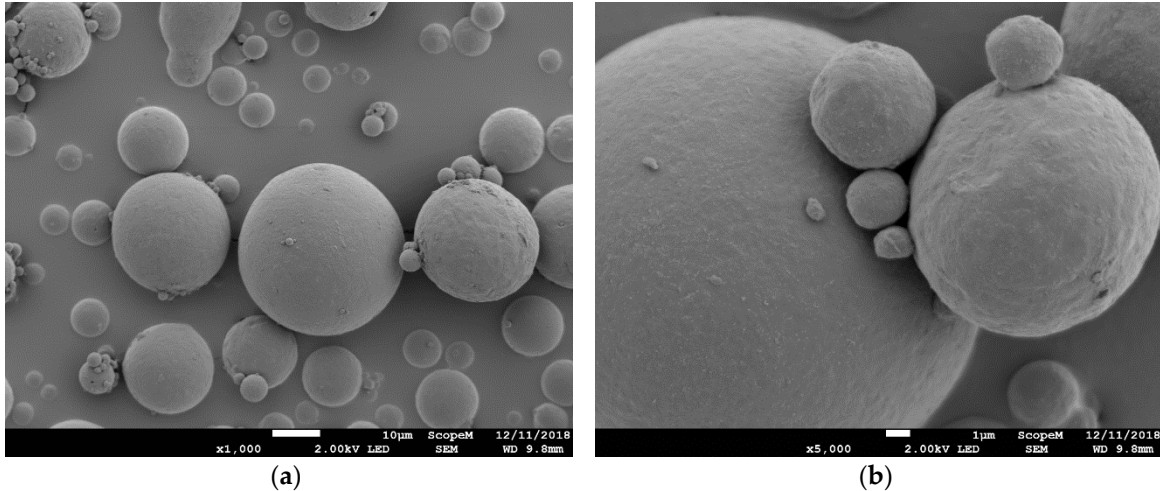

(**a**)                      (**b**)

**Figure 7.** (**a**) Overview of spherical PBT particles; (**b**) Detail of PBT particles.

The particle shape was evaluated quantitatively with optical microscopy. Three parameters can be taken as useful shape descriptors, as elaborated in [21]. The elliptic smoothness $\sigma_E$ is defined as the ratio between the perimeter of the particle and that of its surrounding fitted ellipse. This gives an impression of the regularity of the particle surface. The shape is further defined by the particle solidity S, the ratio between the surface area of the particle and its surrounding convex hull. Finally, the aspect ratio A defines whether a particle is stretched or has a more isotropic shape.

The shape descriptors found for the PBT powder and two reference materials are listed in Table 4. For the PBT powder, a $\sigma_E$ of 1.06 is determined. This corresponds well with successful current laser sintering materials such as Duraform®PA (3DSystems, Rock Hill, SC, United States, 2018) and iCoPP (Asphia PP, Aspect Inc., Tokyo, Japan, 2018), a spherical polypropylene powder [31]. Similarly, the mean aspect ratio found for the PBT powder is 1.22, indicating only slight deviation from isotropic spherical particles. This value is similar to iCoPP, whereas Duraform®PA typically has a higher aspect ratio. The solidity of the spherical materials is similar, the more irregularly shaped Duraform®PA also has a lower solidity.

**Table 4.** Particle shape descriptors for the PBT powder and commercially used reference powders.

| Powder | $\sigma_E$ [-] | A [-] | S [-] |
|---|---|---|---|
| PBT | $1.06 \pm 0.04$ | $1.22 \pm 0.24$ | $0.93 \pm 0.02$ |
| Duraform®PA | $1.06 \pm 0.04$ | $1.58 \pm 0.39$ | $0.87 \pm 0.06$ |
| iCoPP | $1.04 \pm 0.02$ | $1.10 \pm 0.13$ | $0.94 \pm 0.03$ |

These results show that the melt emulsification process is exceptionally suited to produce particles with sizes and shapes that resemble and even surpass commercially used laser sintering materials. As demonstrated in [15–19], the particle shape to a large extent determines the flowability and packing density of the powder, and in this way the density and mechanical properties of produced parts. The favorable shape of the current PBT powder permits the production of dense, mechanically stable parts.

### 3.4. Powder Flowability

In conjunction with the favorable particle shape, the powder flowability shows promising trends as well. Table 5 summarizes the avalanche angle $\alpha_a$, avalanche fractal $f_A$, and Hausner ratio for the PBT powder and the two previously introduced commercially used reference powders.

**Table 5.** Powder flowability results for the PBT powder and commercially used reference powders.

| Powder | $\alpha_A$ [°] | $f_A$ [-] | HR [-] |
|---|---|---|---|
| PBT | 42.05 ± 3.52 | 1.51 ± 0.16 | 1.097 |
| Duraform®PA | 43.76 ± 4.25 | 1.48 ± 0.20 | 1.221 |
| iCoPP | 45.01 ± 6.24 | 1.56 ± 0.26 | 1.236 |

The avalanche angle represents the resistance of the powder to flow, which is mainly due to frictional forces between particles. A lower angle means a powder flows more easily. The standard deviation is a measure for the homogeneity and consistency of flow, where lower values signify a more controlled behavior. The avalanche fractal quantifies the smoothness of the powder surface directly after an avalanche. The Hausner ratio is defined as the ratio between the bulk, free flowing density of the powder and its tapped density. As a general rule, powders with a HR < 1.25 are free flowing, and considered by most authors [13,29,32,33] to be suitable in terms of flowability for laser sintering .

The PBT powder shows comparable behavior to the commercial powders when avalanche angle and fractal are considered. The most striking difference can be seen in the Hausner ratio. The powder packs very well in its bulk state, which is due in part to the spherical particles, but is also related to the high intrinsic density of the material (1.31 g/cm$^3$) relative to the reference powders.

## 3.5. Thermal Properties

One of the key material features that ensure its successful processing by laser sintering is the presence of a wide sintering window, situated between the onset of crystallization, and the onset of melting. The DSC thermograms in Figure 8 show a sufficient window of 24.8 °C width for the starting granulate material. The window is however narrowed considerably when the material is transformed into a powder. The crystallization onset is shifted to higher temperatures by 10.7 °C, resulting in a sintering window that is merely 17.2 °C wide. This could lead to problems during laser sintering processing, because thermal gradients on the powder bed due to inadequate heating can be in the order of 10–15 °C, especially at the higher temperatures required for the processing of PBT.

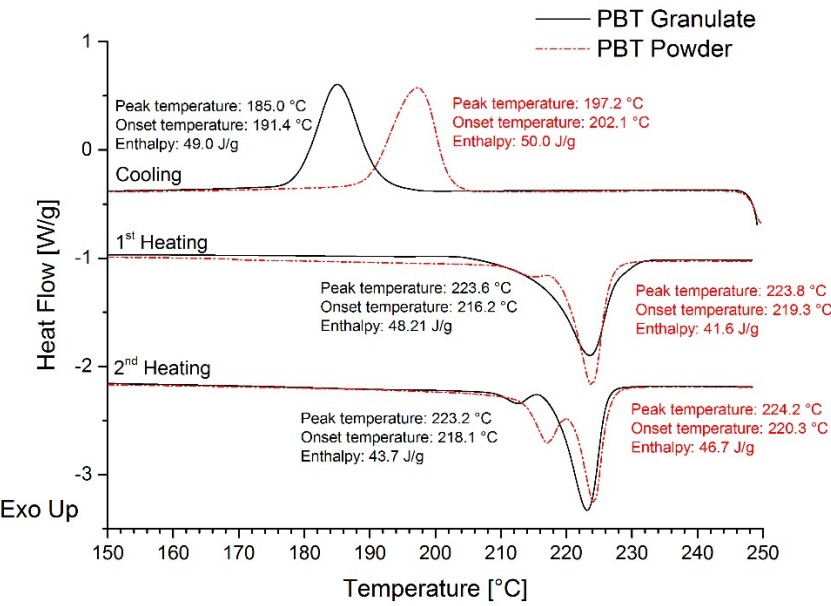

**Figure 8.** Differential scanning calorimetry (DSC) thermograms of PBT granulate (starting material) and produced PBT powder.

It is observed that the change in thermal properties already occurs after extrusion, and remains present as the powder is refined. Arai et al. [13] observed similar behavior in cryogenically milled PBT.

They attributed the shift in crystallization temperature to nucleation by microscopic metal particles introduced to the powder during the milling process. Since the powder in the current research was produced with a different method, it is unlikely that the same mechanism plays a role. Polymer degradation could be an alternative reason, especially considering that a grade without stabilizing additives was used. As it turns out however, even when a commercial, stabilized PBT grade is used, the same shift in crystallization temperature occurs. Results from Pillin et al. [34] on the other hand do support the notion of PBT hydrolysis being a major contributor to the raise in crystallization temperature. It is very well possible that despite stabilizing additives and preliminary drying of the PBT granulate, the water content in the hygroscopic PEG is so high, that degradation via hydrolysis during extrusion cannot be prevented. Additionally, the additives may be washed out during the extrusion process, if they have a higher affinity with the PEG phase.

### 3.6. Rheological Properties

The occurrence of degradation along the process chain from granulate to powder is confirmed by rheological measurements. In Figure 9, the apparent melt viscosity of two PBT grades is shown. The black curves correspond to the PBT 1200M, the red curves stem from the stabilized 1100-S600 PBT, which was transformed into a powder using the same methods as described above and shown in Figure 2. Both grades show considerable reduction of viscosity after having been processed into powder. From the DSC measurements, it appears from the shift in crystallization onset temperature, that degradation already begins during the first process step of melt blending. Further steps including washing and drying however also contribute to degradation. Considering the high surface-to-volume ratio of the small particles and the large amounts of water used, hydrolysis is a substantial problem that must be addressed in future work.

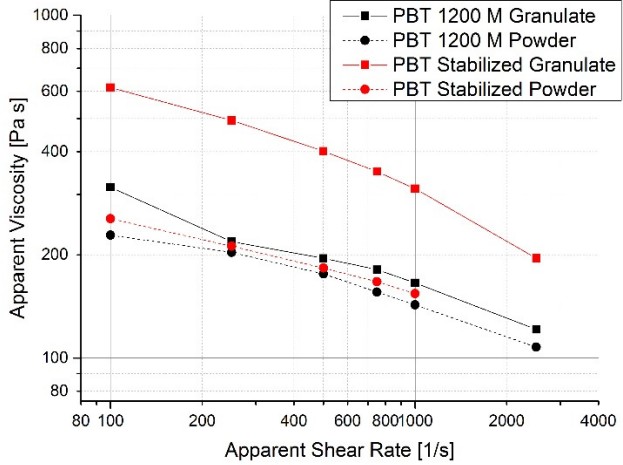

**Figure 9.** Apparent viscosity of PBT granulates and powders at 250 °C.

### 3.7. Laser Sintering

Multiple laser sintering trials were carried out to test the performance of the powder in a commercial laser sintering machine and to find the optimal process parameters. The powder flowability was excellent. The powder could be deposited at process temperature without any issues. The powder bed was smooth and fault-free. The scanned areas of parts formed a clear melt pool, as can be seen in Figure 10a. The narrow sintering window of the material that was already observed in the DSC thermograms caused process issues. Due to a high crystallization rate and associated shrinkage, the parts showed increasing degrees of curling as they were built. Eventually, this caused the parts to be shifted on the powder bed by the recoater, as can be seen in Figure 10b.

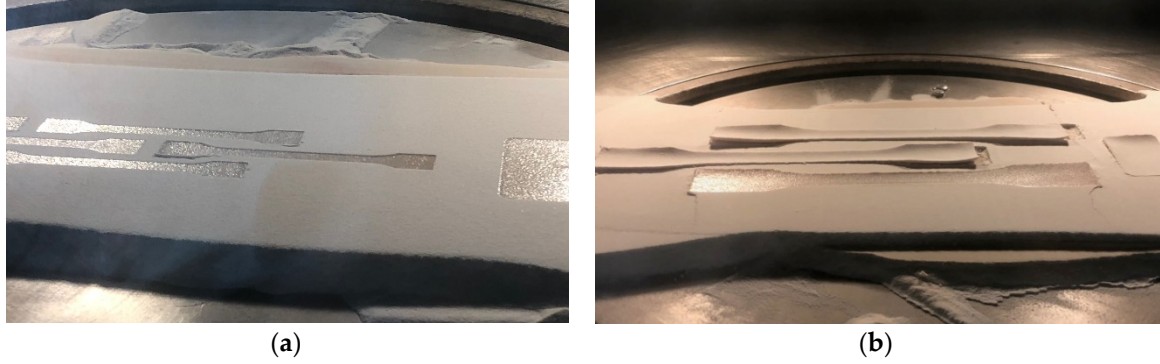

**Figure 10.** (**a**) Sintered tensile bars in powder bed. Minimal curling is visible in the center; (**b**) Parts are moved on the powder bed by the recoater due to curling.

The occurrence of part curling is a clear indication that the powder bed and feed temperatures are too low. Increasing these temperatures however was not possible. As depicted in Figure 1, the powder bed temperature was at the upper limit, ascertain regions started to melt spontaneously, without being scanned by the laser. There were still regions where parts could be built without curling and preliminary melting, but the temperature differences in the machine are too large to accommodate for a material with a sintering window this narrow. In parallel, the feed temperature can be increased to reduce the thermal shock when a new layer with colder powder is deposited on freshly molten surfaces. In this case however, the maximum feed temperature was limited by the fact that small-scale inserts were used. By careful positioning of the parts in locations where the temperature was exactly right so no curling or excess melting occurred, small parts could be built and tested. The parts, shown in Figure 11, are yellowish but otherwise stable and able to withstand sandblasting at mild pressures.

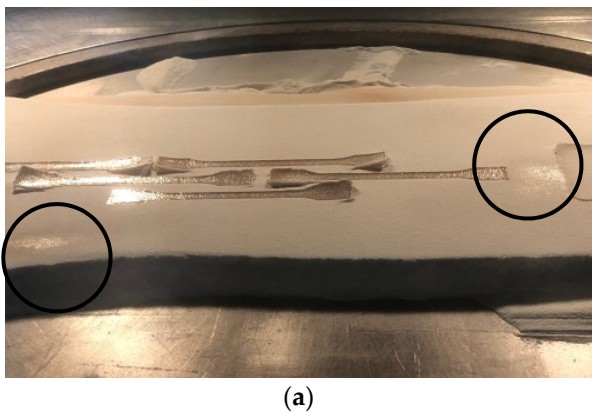
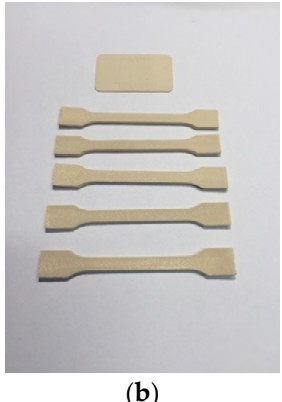

**Figure 11.** (**a**) Glossy hot spots (indicated by the circles) appearing on the powder bed where the temperature is too high; (**b**) Parts that could be built and tested.

The process window of the PBT can be increased if the material is annealed. By annealing the material at 210 °C for 12 hours in a DSC, the onset melting temperature could be increased by almost 8 °C to 227.1 °C. A major disadvantage of annealing however is the limited thermal stability of the material, and the associated thermal degradation in the absence of stabilizers. For the material in its current state, annealing is not a viable option.

*3.8. Tensile Tests*

Three tensile bars were tested. The results are summarized in Table 6. The parts produced with laser sintering exhibit brittle behavior, and suffer considerably in terms of tensile strength ($\sigma_M$) and elongation at break (EaB). It is typical for laser sintering materials to have a higher Young's modulus and much lower elongation at break than their injection molded counterparts, the same was found for

laser sintered PBT parts by other researchers [13]. The drastically reduced tensile strength however indicates the laser sintering process is not the only factor influencing the mechanical properties. The polymer degradation that occurs during the powder production stages, which was confirmed by DSC and capillary rheometry, also plays an important role here.

**Table 6.** Tensile results for PBT tensile bars.

| Material/Method | E [MPa] | $\sigma_M$ [MPa] | EaB [%] |
| --- | --- | --- | --- |
| PBT Powder/SLS | 2211.4 ± 39 | 20.3 ± 2.30 | 0.98 ± 0.1 |
| PBT Granulate/IM | 2280 ± 199 | 52.7 ± 1.66 | 111.8 ± 23.2 |
| PBT Powder/CM | 1900 ± 378 | 23.9 ± 10.5 | 1.1 ± 0.2 |

To compare properties, tensile bars made from different materials and with a variety of processes were tested. A set of specimens made from the unaltered granulate was cut out from an injection molded (IM) sheet. The next set of specimens was made by compression molding sheets with the produced powder as a starting material. The injection molded granulate shows properties corresponding to those reported on the material datasheet. As soon as the granulate is transformed to a powder, the strength and elongation at break deteriorate, regardless of how the specimens were fabricated. This behavior shows that the worsening of properties is not merely the effect of processing by laser sintering, but is mainly due to degradation in the powder fabrication process.

### 3.9. Part Morphology and Microstructure

A cross-section of a tensile bar was investigated with optical microscopy to assess its morphology and microstructure. The bar was produced in a different build job than the bars that were tested in the previous section, but was built with the same scanning parameters and temperatures. Considerable curling of the bar was observed during the build job. As a result, the part started to delaminate, which can clearly be seen in Figure 12. Along the build direction, from bottom to top, the part initially has a dense microstructure. As the build progresses, first signs of delamination can be seen, along with some larger pores. During the last stage, individual layers can be distinguished, alternating with voids containing unmolten particles. Although the curling of the investigated part was more extreme than for the tested tensile bars, it is likely that the microstructure of those bars shows similarities to the transition region shown in Figure 12. This also partly explains the poor mechanical properties.

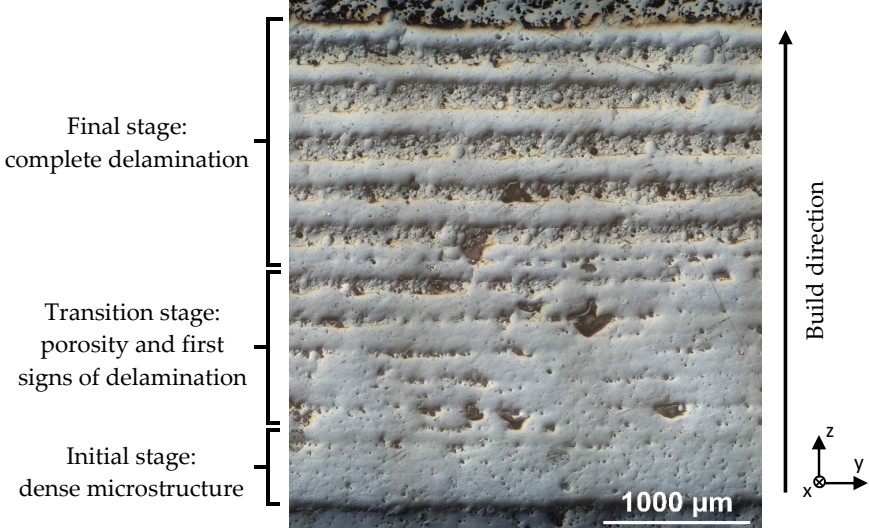

**Figure 12.** Cross-section of a tensile bar. The build (Z) direction runs from bottom to top, the layer application (X) direction is normal to the viewing plane.

## 4. Conclusions

The melt emulsification method is exceptionally suited to make powders with spherical geometry in the size range that is required for laser sintering. Compared to conventional laser sintering powders, the powder that was fabricated in this study showed outstanding flowability and packing density properties. The measured properties could be confirmed through the application of the powder in a commercial laser sintering system.

As a powder production technology for polyesters in general and PBT in particular, melt emulsification however presents critical limitations. It could be demonstrated that the polymer degraded considerably over the course of the powder production process. This initially translated to a narrower process window. For parts that could be built, inferior mechanical properties were found for both laser sintered and compression molded parts made from powder, compared to injection-molded parts made from the starting granulate.

The initial melt emulsification process step in particular leads to degradation, first of all because a PBT grade without heat and hydrolysis stabilization was used in this study. A similar PBT grade with additives showed comparable degradation effects, which may be traced back to the water content in the matrix phase and washing out of stabilizers. The selection of a different water-soluble matrix polymer such as polyvinyl alcohol (PVA) or polyvinylpyrrolidone (PVP) could help reducing hydrolysis during extrusion. Enhanced drying, for example in a circulating hot air dryer would also lead to an improvement. Finally, the addition of more or, other additives can reduce thermal degradation and hydrolysis during all process steps.

This study could validate the viability of the melt emulsification process as a means to produce laser sintering powders with outstanding particle shape, size and flowability. The process is scalable and can be extended to other material systems to allow the production of a wide spectrum of laser sintering powders. The economically viable production of powders on an industrial scale can be realized by combination of individual process steps. Extrusion, washing, and sedimentation can be performed in a single continuous step for example. By such optimizations, the method need not be much more expensive than other powder production processes, such as cryogenic milling. Possible additional costs for producing the powder with melt emulsification can be offset by the benefit of obtaining powders with spherical particles.

**Author Contributions:** R.G.K. conducted the trials, and wrote the paper. M.S. initiated the project, assisted with planning trials, and interpretation of results. K.W. is supervisor of the group, initiated the idea of producing new powders for laser sintering, and gave valuable input on all steps of the study progress.

**Funding:** This project was financially supported by the Swiss Commission for Technology and Innovation (CTI) under project grant nr. 19194.1.

**Acknowledgments:** Martin Rohrer of Geberit AG is gratefully acknowledged for his assistance with blend preparation, tensile testing, and rheological measurements. The authors also wish to extend their gratitude to Gerhard Gielenz of Geberit AG for his invaluable feedback and contributions to this work. Luiz Morales of the Scientific Center for Optical and Electron Microscopy (ScopeM) at ETH Zürich is acknowledged for his assistance with acquiring SEM images.

**Conflicts of Interest:** The authors declare no conflict of interest. The funders had no role in the design of the study; in the collection, analyses, or interpretation of data; in the writing of the manuscript, or in the decision to publish the results.

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
