# Peer review of "Production and Processing of a Spherical Polybutylene Terephthalate Powder for Laser Sintering"

_applsci, doi:10.3390/app9071308_

Round 1
Reviewer 1 Report
The paper describes an interesting route to produce polymer powder for laser sintering. It omits the annoying step of cry-milling. In that sense, the paper merits publication but a few comments need to be taken into account:
the manuscript contains a lot (too many) self-citations, seeming to indicate that the authors are the only ones working on powder flow for SLS. This is not true and some other important work has been done by other groups (e.g. the Exeter group, Nottingham group, Leuven group, ...) and this should be acknowledged in the manuscript;
there seems to be a lot of changes in the thermal properties of the PBT after the various treatments. Did the authors consider the polymorphic nature of the material and did they test the crystallographic properties of the material?
drying of PBT might effect rheological properties to a large extent. Has this been checked?
a lot is attributed to the degradation of the PBT. Can the authors support this by TGA experiments or by molecular weight determinations to check potential hydrolysis or other chemical effects?
the method is cumbersome and time consuming, it would be worthwhile to include an economical assessment of the proposed method as compared to the cry milling.
Author Response
The authors thank the reviewer for reviewing the manuscript. Where possible and applicable, the manuscript was changed. Please refer to the answers below for our detailed comments.
- The authors agree with the reviewer concerning the work done by other groups and have added references to their work to better reflect the state of the art.
- The authors are aware of the polymorphic nature of PBT. In terms of thermal properties, this was not investigated in detail however. The main effect that was observed in DSC thermograms, was the increase of crystallization temperature for the PBT powder. Even though it is possible that different crystalline modifications form during crystallization at higher temperatures, they can’t be the reason the material starts to crystallize earlier. Rather, this has to do with increased chain mobility (reflected by the reduced melt viscosity measured for the powder), and possibly with nucleation. It would be interesting to investigate the crystalline structure of built parts. Since crystallization in the SLS process occurs very slowly, it is likely that the material has a different crystalline modification. This might also partly explain the brittleness of laser sintered samples.
- The apparent viscosity of the materials was only checked before and after the powder production process, not after each individual process step. The authors agree with the reviewer that it is very likely the drying has a large influence on the polymer rheology. The prolonged time at high temperature and humidity can have drastic effects on the polymer molecular weight and with that, its rheology. Improvement of the drying process is therefore one of the recommendations given in the conclusions.
- The molecular weight of the PBT was not directly measured. However, by measurement of the apparent viscosity, reduction of molecular weight can be indirectly inferred. Other signs are yellowing of the powder and parts, and the brittleness of parts made from powder by compression molding. Determination of the exact nature of the polymer degradation was not the focus of this study. However, Oxidation Induction Temperature (OIT) measurements were performed to test the polymer stability. Since these measurements could not be performed reproducibly, and therefore provided inconclusive results, they were not reported.
- The authors agree with the reviewer that the economic feasibility of the melt emulsification process should be assessed. The estimations of the authors have been added to the text. The authors also wish to emphasize that the reported work was carried out with available laboratory equipment, which was not specifically optimized for the process. To implement the process commercially, many steps can be combined and improved. The extrusion, washing, and sedimentation could be performed in a single continuous step, for example. The drying step can’t be skipped, but is sometimes also necessary for cryogenically ground materials, as is sieving of the resulting powder. And even though cryogenic grinding has the reputation for being low-cost, liquid nitrogen doesn’t come cheap either.

Reviewer 2 Report
This manuscript describes the production of a spherical polybutylene terephthalate (PBT) powder and its processing with selective laser sintering (SLS). In the current investigation was evaluated the correlation between extruder rotational speed and nozzle diameter on the particle size. The powder was then characterized from a morphological, thermal and flowability point of view. Finally, the mechanical properties of the 3D printed part via SLS was compared to those obtained via compression and injection molding. The manuscript is well written, the content of this study is interesting and the results are useful for the researchers in the related field. The experimental part is very accurate and the powder has been really fine characterize from morphological and mechanical point of view and the conclusion are totally supported from experimental data. Moreover, a really accurate flowability study has been conducted and the scientific soundness of the manuscript result very high. However, some doubts are correlated to the low mechanical properties of the sintered object and the size of the powder results too big for the sintering parameters used during the process. I kindly suggest the acceptance of the present manuscript in Applied Science after some minor revision:
1. Please provide a better explanation of Figure 3 and D10, D50 and D90 in the text because they result a little hard to understand;
2. If the layer thickness used is 0.12 mm a particle size of 100µm results too large and could act as a defect of the layer. Maybe using particle with smaller size could increase the mechanical properties of the printed part;
3. Please provide a morphological characterization of the 3D printed sample in order to have more information about the sintered material internal morphology. Maybe the analysis could better explain the low mechanical behaviour of the object obtained from the powder;
4. Figure 5 report 6 curves related to the size distribution of PBT powder after different process step. However the legend only report 3 curves. Please correct the legend of the figure;
5. Paragraph 3.7 clearly shows that the author faced many problems during the sintering process due to high degrees of curling of the printed object. However, it is not clear how the author solve the problem without changing the temperature of the chamber or the power of the laser. Please give a better explanation about how you printed the object avoiding the curling effect.
Author Response
The authors thank the reviewer for reviewing the manuscript. Where possible and applicable, the manuscript was changed. Please refer to the answers below for our detailed comments.
- An additional explanation regarding figure 3 has been added to the text.
- It is true that the particle size at a certain point limits the layer thickness. The authors believe this not to be the case for the PBT powder. The D90 of the size distribution is with 110 µm still smaller than the layer thickness of 120 µm. Moreover, where parts are built, the surface of the melt pool goes down with respect to the powder surface, leaving more space to deposit particles. Depending on the width of the distribution, the packing density of the powder bed can be increased. The authors agree with the reviewer that the size distribution can be optimized further to improve packing density. However, this by itself is an extensive research work that goes beyond the scope of this study.
- The morphology and porosity of a polished cross-section of a part was evaluated with optical microscopy. The results and evaluations have been added to the manuscript.
- The caption and legend of Figure 5 have been updated
- The text has been adapted to better reflect the changes that were made. Ultimately, one has to carefully choose the locations to build a part where the powder bed temperature is exactly right. Parts may then be built without curling and without premature powder bed melting.

Round 2
Reviewer 1 Report
-